# DropIT: Dropping Intermediate Tensors for Memory-Efficient DNN Training

**Joya Chen[1*], Kai Xu[1*], Yuhui Wang[1], Yifei Cheng[2], Angela Yao[1]**
[1]National University of Singapore
[2]University of Science and Technology of China

joyachen@u.nus.edu   {kxu,yuhuiw,ayao}@comp.nus.edu.sg   chengyif@mail.ustc.edu.cn

## Abstract

A standard hardware bottleneck when training deep neural networks is GPU memory. The bulk of memory is occupied by caching intermediate tensors for gradient computation in the backward pass. We propose a novel method to reduce this footprint - Dropping Intermediate Tensors (DropIT). DropIT drops min-$k$ elements of the intermediate tensors and approximates gradients from the sparsified tensors in the backward pass. Theoretically, DropIT reduces noise on estimated gradients and therefore has a higher rate of convergence than vanilla-SGD. Experiments show that we can drop up to 90% of the intermediate tensor elements in fully-connected and convolutional layers while achieving higher testing accuracy for Visual Transformers and Convolutional Neural Networks on various tasks (*e.g.*, classification, object detection, instance segmentation). Our code and models are available at https://github.com/chenjoya/dropit.

## 1 Introduction

The training of state-of-the-art deep neural networks (DNNs) (Krizhevsky et al., 2017; Simonyan & Zisserman, 2015; He et al., 2016; Vaswani et al., 2017; Dosovitskiy et al., 2021) for computer vision often requires a large GPU memory. For example, training a simple visual transformer detection model ViTDet-B (Li et al., 2022), with its required input image size of $1024 \times 1024$ and batch size of 64, requires ~700 GB GPU memory. Such a high memory requirement makes the training of DNNs out of reach for the average academic or practitioner without access to high-end GPU resources.

When training DNNs, the GPU memory has six primary uses (Rajbhandari et al., 2020): network parameters, parameter gradients, optimizer states (Kingma & Ba, 2015), intermediate tensors (also called activations), temporary buffers, and memory fragmentation. Vision tasks often require training with large batches of high-resolution images or videos, which can lead to a significant memory cost for intermediate tensors. In the instance of ViTDet-B, approximately 70% GPU memory cost (~470 GB) is assigned to the intermediate tensor cache. Similarly, for NLP, approximately 50% of GPU memory is consumed by caching intermediate tensors for training the language model GPT-2 (Radford et al., 2019; Rajbhandari et al., 2020). As such, previous studies (Gruslys et al., 2016; Chen et al., 2016; Rajbhandari et al., 2020; Feng & Huang, 2021) treat the intermediate tensor cache as the largest consumer of GPU memory.

For differentiable layers, standard implementations store the intermediate tensors for computing the gradients during back-propagation. One option to reduce storage is to cache tensors from only some layers. Uncached tensors are recomputed on the fly during the backward pass – this is the strategy of gradient checkpointing (Gruslys et al., 2016; Chen et al., 2016; Bulo et al., 2018; Feng & Huang, 2021). Another option is to quantize the tensors after the forward computation and use the quantized values for gradient computation during the backward pass (Jain et al., 2018; Chakrabarti & Moseley, 2019; Fu et al., 2020; Evans & Aamodt, 2021; Liu et al., 2022) – this is known as activation compression training (ACT). Quantization can reduce memory considerably, but also brings inevitable performance drops. Accuracy drops can be mitigated by bounding the error at each layer through adaptive quantization (Evans & Aamodt, 2021; Liu et al., 2022), *i.e.* adaptive ACT. However, training time consequently suffers as extensive tensor profiling is necessary during training.

---

*Equal contribution.

In this paper, we propose to reduce the memory usage of intermediate tensors by simply dropping elements from the tensor. We call our method Dropping Intermediate Tensors (DropIT). In the most basic setting, dropping indices can be selected randomly, though dropping based on a min-$k$ ranking on the element magnitude is more effective. Both strategies are much simpler than the sensitivity checking and other profiling strategies, making DropIT much faster than adaptive ACT.

During training, the intermediate tensor is transformed over to a sparse format after the forward computation is complete. The sparse tensor is then recovered to a general tensor during backward gradient computation with dropped indices filled with zero. Curiously, with the right dropping strategy and ratio, DropIT has improved convergence properties compared to SGD. We attribute this to the fact that DropIT can, theoretically, reduce noise on the gradients. In general, reducing noise will result in more precise and stable gradients. Experimentally, this strategy exhibits consistent performance improvements on various network architectures and different tasks.

To the best of our knowledge, we are the first to propose activation sparsification. The closest related line of existing work is ACT, but unlike ACT, DropIT leaves key elements untouched, which is crucial for ensuring accuracy. Nevertheless, DropIT is orthogonal to activation quantization, and the two can be combined for additional memory reduction with higher final accuracy. The key contributions of our work are summarized as follows:

- We propose DropIT, a novel strategy to reduce the activation memory by dropping the elements of the intermediate tensor.

- We theoretically and experimentally show that DropIT can be seen as a noise reduction on stochastic gradients, which leads to better convergence.

- DropIT can work for various settings: training from scratch, fine-tuning on classification, object detection, etc. Our experiments demonstrate that DropIT can drop up to 90% of the intermediate tensor elements in fully-connected and convolutional layers with a testing accuracy higher than the baseline for CNNs and ViTs. We also show that DropIT is much better regarding accuracy and speed compared to SOTA activation quantization methods, and it can be combined with them to pursue higher memory efficiency.

## 2 RELATED WORK

**Memory-efficient training.** Current DNNs usually incur considerable memory costs due to huge model parameters (*e.g.* GPTs (Radford et al., 2019; Brown et al., 2020)) or intermediate tensors (*e.g.* , high-resolution feature map (Sun et al., 2019; Gu et al., 2022)). The model parameters and corresponding optimizer states can be reduced with lightweight operations (Howard et al., 2017; Xie et al., 2017; Zhang et al., 2022), distributed optimization scheduling (Rajbhandari et al., 2020), and mixed precision training (Micikevicius et al., 2018). Nevertheless, intermediate tensors, which are essential for gradient computation during the backward pass, consume the majority of GPU memory (Gruslys et al., 2016; Chen et al., 2016; Rajbhandari et al., 2020; Feng & Huang, 2021), and reducing their size can be challenging.

**Gradient checkpointing.** To reduce the tensor cache, gradient checkpointing (Chen et al., 2016; Gruslys et al., 2016; Feng & Huang, 2021) stores tensors from only a few layers and recomputes any uncached tensors when performing the backward pass; in the worst-case scenario, this is equivalent to duplicating the forward pass, so any memory savings come as an extra computational expense. InPlace-ABN (Bulo et al., 2018) halves the tensor cache by merging batch normalization and activation into a single in-place operation. The tensor cache is compressed in the forward pass and recovered in the backward pass. Our method is distinct in that it does not require additional recomputation; instead, the cached tensors are sparsified heuristically.

**Activation compression.** (Jain et al., 2018; Chakrabarti & Moseley, 2019; Fu et al., 2020; Evans & Aamodt, 2021; Chen et al., 2021; Liu et al., 2022) explored lossy compression on the activation cache via low-precision quantization. (Wang et al., 2022) compressed high-frequency components while (Evans et al., 2020) adopted JPEG-style compression. In contrast to all of these methods, DropIT reduces activation storage via sparsification, which has been previously unexplored. In addition, DropIT is more lightweight than adaptive low-precision quantization methods (Evans & Aamodt, 2021; Liu et al., 2022).

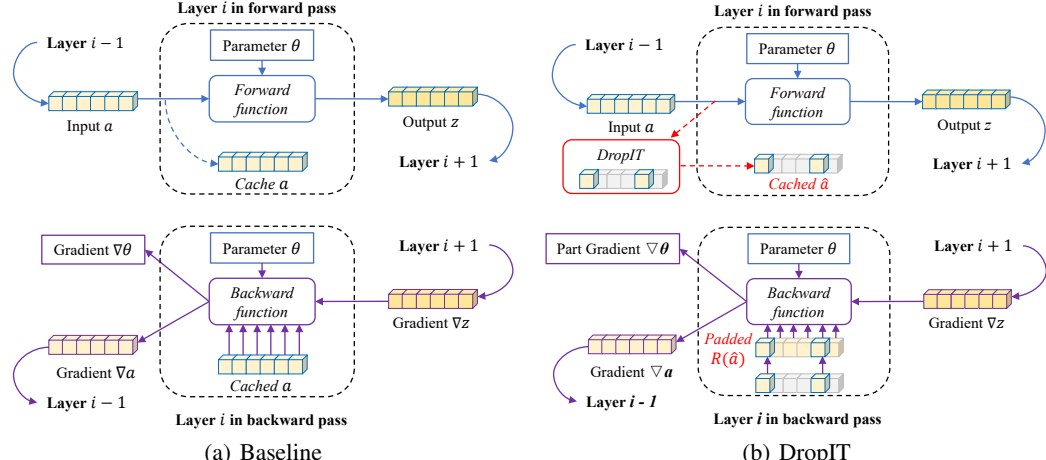

Figure 1: For a regular baseline network (a), the intermediate tensor is fully cached in the forward pass to be used for gradient computation during the backward pass. For DropIT, elements of the intermediate tensors are dropped during caching; only the retained elements with zero padding are used for gradient computation during the backward pass. DropIT can save GPU memory for two reasons. First, cached tensors are accumulated layer by layer during the forward pass, and DropIT sparsifies them, thereby reducing maximum memory allocation. Second, backward tensors are released after use, making the memory cost associated with padding negligible. Best viewed in color.

**Gradient approximation.** Approximating gradients has been explored in large-scale distributed training to limit communication bandwidth for gradient exchange. (Strom, 2015; Dryden et al., 2016; Aji & Heafield, 2017; Lin et al., 2018) propose dropping gradients based on some fixed thresholds and sending only the most significant entries of the stochastic gradients with the guaranteed convergence (Stich et al., 2018; Cheng et al., 2022; Chen et al., 2020). Instead of dropping gradient components, DropIT directly drops elements within intermediate tensors as our objective is to reduce the training memory.

## 3    METHODOLOGY

### 3.1    PRELIMINARIES

We denote the forward function and learnable parameters of the $i$-th layer as $l$ and $\theta$, respectively. In the forward pass, $l$ operates on the layer's input $a$ to compute the output $z$: [1]

$$z = l(a, \theta). \tag{1}$$

For example, if layer $i$ is a convolution layer, $l$ would indicate a convolution operation with $\theta$ representing the kernel weights and bias parameter.

Given a loss function $F(\Theta)$, where $\Theta$ represents the parameters of the entire network, the gradient, with respect to $\theta$ at layer $i$, can be estimated according to the chain rule as

$$\nabla\theta \triangleq \frac{\partial F(\Theta)}{\partial \theta} = \nabla z \frac{\partial z}{\partial \theta} = \nabla z \frac{\partial l(a, \theta)}{\partial \theta}, \tag{2}$$

where $\nabla z \triangleq \frac{\partial F(\Theta)}{\partial z}$ is the gradient passed back from layer $i + 1$. Note that the computation of $\frac{\partial l(a,\theta)}{\partial \theta}$ requires $a$ if the forward function $l$ involves tensor multiplication between $a$ and $\theta$. This is the case for common learnable layers, such as convolutions in CNNs and fully-connected layers in transformers. As such, $a$ is necessary for estimating the gradient and is cached after it is computed

---

[1]Note that the output from the previous layer $i-1$, i.e. $a^i = z^{i-1}$. However, we assign different symbols to denote the input and output of a given layer explicitly; this redundant notation conveniently allows us, for clarity purposes, to drop the explicit reference of the layer index $i$ as a superscript.

| Layer Type | Parameter $\theta$ | Tensor $a$ |
|---|---|---|
| Convolution | $O(C_a C_z K^2)$ | $O(BC_a L_a)$ |
| Fully Connected | $O(C_a C_z)$ | $O(BL_a C_a)$ |

Table 1: Space complexity for parameters and intermediate tensors in a single layer. $B$: batch size, $L_a$: input sequence length (*e.g.*, width×height), $C_a, C_z$: the number of input, output channels, $K$: convolutional kernel size. Typically, $C_a, C_z, K$ would be fixed once the model has been built, so the complexity for intermediate tensors would be considerable with large $B, L_a$.

during the forward pass, as illustrated in Figure 1(a). A common way to reduce storage for $a$ is to store a quantized version (Jain et al., 2018; Chakrabarti & Moseley, 2019; Fu et al., 2020; Evans & Aamodt, 2021; Liu et al., 2022). Subsequent gradients in the backward pass are then computed using the quantized $a$. The gradient $\nabla a$ can be estimated similarly via chain rule as

$$\nabla a \triangleq \frac{\partial F(\Theta)}{\partial a} = \nabla z \frac{\partial z}{\partial a} = \nabla z \frac{\partial l(a,\theta)}{\partial a}. \tag{3}$$

Analogous to Eq. 2, the partial $\frac{\partial l(a,\theta)}{\partial a}$ may depend on the parameter $\theta$ and $\theta$ is similarly stored in the model memory. However, the stored $\theta$ always shares memory with the model residing in the GPU, so it does not incur additional memory consumption. Furthermore, $\theta$ typically occupies much less memory. In Table 1, the intermediate tensor's space complexity becomes significant when $B$ or $L_a$ is large, which is common in CV and NLP tasks.

## 3.2 DROPPING INTERMEDIATE TENSORS

Let $\mathcal{X}$ denote the set of all indices for an intermediate tensor $a$. Suppose that $\mathcal{X}$ is partitioned into two disjoint sets $\mathcal{X}_d$ and $\mathcal{X}_r$, *i.e.* $\mathcal{X}_r \cap \mathcal{X}_d = \emptyset$ and $\mathcal{X}_r \cup \mathcal{X}_d = \mathcal{X}$. In DropIT, we introduce a dropping operation $D(\cdot)$ to sparsify $a$ into $\hat{a}$, where $\hat{a}$ consists of the elements $a_{\mathcal{X}_r}$ and the indices $\mathcal{X}_r$, *i.e.* $\hat{a} = D(a) = \{a_{\mathcal{X}_r}, \mathcal{X}_r\}$. The sparse $\hat{a}$ can be used as a substitute for $a$ in Eq. 2. While sparsification can theoretically reduce both storage and computation time, we benefit only from storage savings in practice. We retain general matrix multiplication because the sparsity rate is insufficient for sparse matrix multiplication to provide meaningful computational gains. As such, the full intermediate tensors are recovered for gradient computation, *i.e.* $\nabla\theta \approx \nabla z \frac{\partial l(R(\hat{a}),\theta)}{\partial \theta}$, where $R(\cdot)$ represents the process that inflates $\hat{a}$ back to a general matrix with dropped indices filled with zero. The overall procedure is demonstrated in Figure 1(b).

Consider for a convolutional layer with $C_z$ kernels of size $K \times K$. For the $j^{\text{th}}$ kernel, where $j \in [1, C_z]$, the gradient at location $(u, v)$ for the $k^{\text{th}}$ channel is given by convolving incoming gradient $\nabla z$ and input $a$:

$$\nabla\theta_{j,k}(u,v) = \sum_{(n,x,y)\in\mathcal{X}} \nabla z_j^n(x,y)a_k^n(x',y'), \tag{4}$$

where $x' = x + u$ and $y' = y + v$. The set $\mathcal{X}$ in this case would denote the set of all sample indices $n \in [1, B]$ and all location indices $(x, y) \in [1, W] \times [1, H]$ in the feature map. Without any loss in generality, we can partition $\mathcal{X}$ into two disjoint sets $\mathcal{X}_r$ and $\mathcal{X}_d$ to split Eq. 4 as

$$\nabla\theta_{j,k}(u,v) = \left[ \sum_{(n,x,y)\in\mathcal{X}_d} \nabla z_j^n(x,y)a_k^n(x',y') + \underbrace{\sum_{(n,x,y)\in\mathcal{X}_r} \nabla z_j^n(x,y)\, a_k^n(x',y')}_{\top\theta_{j,k}(u,v)} \right]. \tag{5}$$

Assume now, that some element $a_k^n(x',y')$ is small or near-zero; in CNNs and Transformers, such an assumption is reasonable due to preceding batch/layer normalization and ReLU or GeLU activations (see Figure 3). Accordingly, this element's contribution to the gradient will also be correspondingly small. If we assign the spatial indices $(x,y)$ in sample $n$ of all small or near-zero elements to $\mathcal{X}_d$, then we can approximate the gradient $\nabla\theta_{j,k}(u,v)$ with simply the second term of Eq. 5. We denote the approximated gradient as $g_{dropit} = \top\theta_{j,k}(u,v)$.

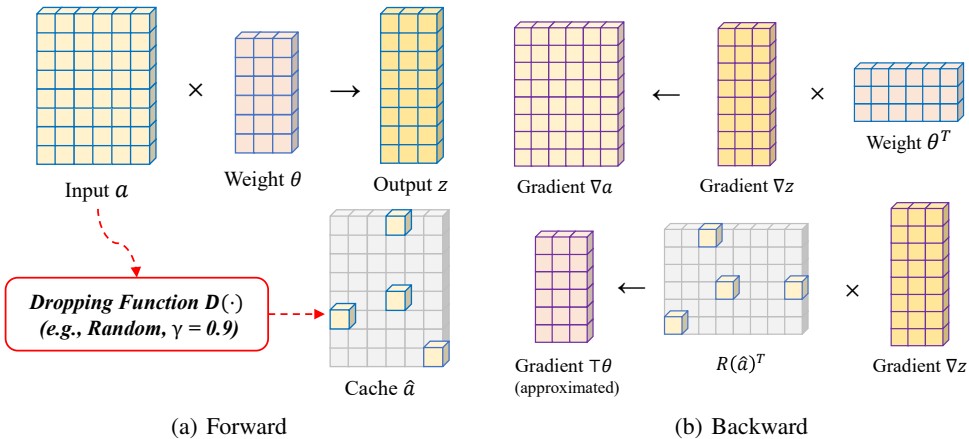

(a) Forward  (b) Backward

Figure 2: Forward and backward of DropIT on the fully-connected layer (without bias). In the forward pass, we sparsify the cache tensor and drop $\gamma$ percentage storage. In the backward pass, only saved elements participate in the gradient computation.

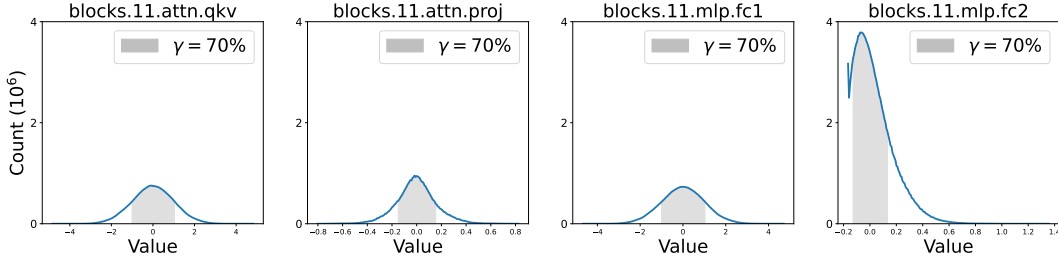

Figure 3: Distribution of element values in intermediate tensors' on DeiT-Ti. Dropped elements are shaded in grey. DropIT with min-$k$ only discards elements that are close to zero. Here we only show the final block while observing that the distributions of other blocks are similar.

For a fully connected layer, the approximated gradient can be defined similarly as

$$g_{dropit} = \top\theta_{j,k} = \sum_{n \in \mathcal{X}_r} \nabla z_j^n \, a_k^n. \tag{6}$$

A visualization of the gradient approximation is shown in Figure 2. With the approximated gradient, we can use any standard deep learning optimization scheme to update the parameters.

### 3.3  Dropping Function $D(\cdot)$

We define the overall dropping rate as $\gamma = \frac{|\mathcal{X}_r|}{BC_a}$ for a fully connected layer and $\gamma = \frac{|\mathcal{X}_r|}{BC_a HW}$ for a convolutional layer. $\gamma$ can be varied and will be used later to define the dropping function $D(\cdot)$. As we aim to drop elements with minimal contribution to the gradient, it is logical to perform a min-$k$ based selection on the elements' magnitudes before dropping the elements. As a baseline comparison, we also select $\mathcal{X}_d$ based on uniform random sampling. We investigate the following options for $D(\cdot)$:

**Random Elements**: $\gamma$ fraction of elements are dropped randomly within a mini-batch.

**Min-K Elements**: Within a mini-batch, we drop the smallest $\gamma$ fraction of elements according to their absolute magnitudes.

### 3.4 THEORETICAL ANALYSIS

Below, we analyze convergence for dropping min-$k$ elements. The gradient of Stochastic Gradient Descent (SGD) is commonly viewed as Gradient Descent (GD) with noise:

$$g_{sgd} = g_{gd} + n(0, \xi^2), \tag{7}$$

where $n$ represents some zero-mean noise distribution with a variance of $\xi^2$ introduced by variation in the input data batches.

With min-$k$ dropping, the gradient becomes biased; we assume it can be modeled as:

$$g_{min\text{-}k} = \alpha g_{gd} + \beta n(0, \xi^2). \tag{8}$$

That is, min-$k$ dropping results in a bias factor $\alpha$ while affecting noise by a factor of $\beta$. $\alpha$ and $\beta$ vary each iteration, i.e., $\alpha = \{\alpha_1, \alpha_2, ..., \alpha_t\}$ and $\beta = \{\beta_1, \beta_2, ..., \beta_t\}$. Additionally in Appendix A.2, we provide a nonlinear approximation to $g_{min-k}$ that achieves same convergence.

By scaling the learning rate with a factor of $\frac{1}{\alpha}$, the gradient after min-$k$ dropping as given in Eq. 8 can also be expressed as:

$$g_{min\text{-}k} = g_{gd} + \frac{\beta}{\alpha} n(0, \xi^2). \tag{9}$$

We can formally show (see Appendix A.3) that $\mathbb{E}[\alpha] \geq \mathbb{E}[\beta] \geq 1 - \gamma$ and therefore $\mathbb{E}[\frac{\beta}{\alpha}] \leq 1$. This suggests that min-$k$ dropping reduces the noise of the gradient. With less noise, better theoretical convergence is expected.

Similar to convergence proofs in most optimizers, we will assume that the loss function $F$ is $L$-smooth. Under the $L$-smooth assumption, for SGD with a learning rate $\eta$ and min-$k$ dropping with a learning rate $\frac{\eta}{\alpha_t}$, we can reach the following convergence after $T$ iterations:

$$\text{SGD:} \quad \frac{1}{T}\mathbb{E}\sum_{t=1}^{T} \|\nabla F(x_t)\|^2 \leq \frac{2(F(x_1) - F(x^*))}{T\eta} + \eta L \xi^2 \tag{10}$$

$$\text{DropIT with min-}k\text{:} \quad \frac{1}{T}\mathbb{E}\sum_{t=1}^{T} \|\nabla F(x_t)\|^2 \leq \frac{2(F(x_1) - F(x^*))}{T\eta} + \eta L \xi^2 \frac{1}{T}\sum_{t=1}^{T} \frac{\beta_t^2}{\alpha_t^2}, \tag{11}$$

where $x^*$ indicates an optimal solution. Full proof can be found in Appendix A.1. Note that the two inequalities differ only by the second term in the right-hand side. $\alpha_t$ represents the bias caused by dropping at the $t$-th iteration and $\beta_t$ measures the noise reduction effect after dropping. We further investigate $\alpha$ and $\beta$ in the supplementary and show that under certain conditions, $\mathbb{E}[\alpha] \geq \mathbb{E}[\beta]$, thereby reducing the noise and improving the convergence of DropIT from standard SGD.

### 3.5 DROPIT FOR NETWORKS

For some layers, *e.g.* normalization and activations, $\frac{\partial l(a,\theta)}{\partial a}$ may also depend on $a$. In these cases, we do not drop the cache of intermediate tensors as this will affect subsequent back-propagation. For DropIT, dropping happens only when the gradient flows to the parameters, which prevents the aggregation of errors from approximating the gradient.

Now, we have discussed dropping tensor elements from the cache of a single layer. DropIT is theoretically applicable for all convolutional and fully-connected layers in a network since it does not affect the forward pass. For Visual Transformers (Dosovitskiy et al., 2021), we apply DropIT for most learnable layers, though we ignore the normalization and activations like LayerNorm (Ba et al., 2016) and GELU (Hendrycks & Gimpel, 2016)). The applicable layers include fully-connected layers in multi-head attention and MLPs in each block, the beginning convolutional layer (for patches projection), and the final fully-connected classification layer. For CNNs the applicable layers include all convolutional layers and the final fully-connected classification layer. We leave networks unchanged during inference.

| Strategy | Dropping Rate $\gamma$ | | | | | | | | | |
|---|---|---|---|---|---|---|---|---|---|---|
| | 0%(*Baseline*) | 10% | 20% | 30% | 40% | 50% | 60% | 70% | 80% | 90% |
| *Random* | 72.1* | **72.4** | **72.4** | 72.0 | 71.7 | 70.8 | 69.6 | 68.1 | 65.8 | 60.8 |
| *Min-K* | 72.1* | **72.1** | **72.1** | **72.2** | **72.4** | **72.5** | **72.4** | **72.1** | 70.8 | 66.4 |

*\* From Touvron et al. (2021)'s official implementation, we obtain 72.13 with public weights and our training.*

Table 2: Ablation study on dropping strategy and dropping rate. Reported results are top-1 accuracy on the ImageNet-1k validation set, achieved by DeiT-Ti training from scratch on the ImageNet-1k training set. We highlight that the accuracy is higher than baseline ( $\geq 72.1$).

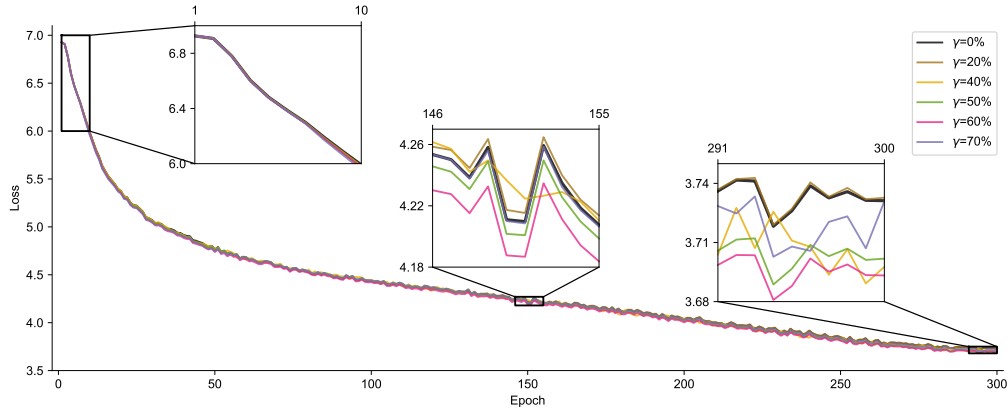

Figure 4: Training loss curves of Min-K DropIT. Baseline ($\gamma = 0\%$) is bolded. $\gamma = 80\%, 90\%$ are hidden as their losses are obviously higher than the baseline. $\gamma = 10\%, 30\%$ are also hidden for easier viewing. $\gamma = 40\% \sim 70\%$ achieve lower loss than baseline at the end. Best viewed in color.

## 4 EXPERIMENTS

In this section, we present a comprehensive evaluation of DropIT's effectiveness, leveraging experiments on training from scratch on ImageNet-1k (Russakovsky et al., 2015). Our results demonstrate that DropIT outperforms existing methods by achieving lower training loss, higher testing accuracy, and reduced GPU memory consumption. We showcase the versatility of DropIT in various fine-tuning scenarios, such as ImageNet-1k to CIFAR-100 (Krizhevsky et al., 2009), object detection, and instance segmentation on MS-COCO (Lin et al., 2014). Furthermore, we compare DropIT with recent state-of-the-art ACT methods (Pan et al., 2022; Liu et al., 2022) and establish its superiority in terms of accuracy, speed, and memory cost.

### 4.1 EXPERIMENTAL DETAILS

**Models.** For image classification, we employed DeiT (Touvron et al., 2021) instead of vanilla ViT (Dosovitskiy et al., 2021) since it doesn't require fine-tuning from ImageNet-21k. DeiT and ViT share the same architecture, differing only in their training hyper-parameters. Additionally, for transfer learning, we utilized Faster/Mask R-CNN models (Ren et al., 2017; He et al., 2017) to evaluate our approach in object detection and instance segmentation.

**Implementation Details.** We use the official implementations of DeiT (without distillation) and Faster/Mask R-CNN, and keep all hyper-parameters consistent. The only difference is that we compute gradients using DropIT. Our implementation is based on PyTorch 1.12 (Paszke et al., 2019), and we utilize torch.autograd package. During the forward pass, we use DropIT to convert the dense tensor to coordinate format, and recover it during the backward pass. The min-$k$ strategy of DropIT is implemented by torch.topk, which retains elements with the largest absolute value, based on a proportion of $1-\gamma$. The corresponding indices of these elements are also maintained. For all experiments, we follow DeiT (Touvron et al., 2021) and set a fixed random seed of 0. We measure training speed and memory on NVIDIA RTX A5000 GPUs. Additional details can be found in Appendix A.8.

(a) CIFAR-100 fine-tuning results. DeiT networks are initialized from their publicly available pre-trained ImageNet-1k weights.

| Network | DropIT | Top-1 Accuracy |
|---------|--------|----------------|
| *DeiT-S* | - | 89.7 |
|          | $\gamma = 90\%$ | **90.1** |
| *DeiT-B* | - | 90.8 |
|          | $\gamma = 90\%$ | **91.3** |

(b) Detection & instance segmentation on COCO. *R50-FPN* denotes ResNet-50 with FPN (Lin et al., 2017), initialized from public ImageNet-1k weights.

| Network | DropIT | $AP^{box}$ | $AP^{mask}$ |
|---------|--------|-----------|------------|
| *Faster R-CNN* | - | 37.0 | n/a |
| *(R50-FPN)* | $\gamma = 90\%$ | **37.2** | n/a |
| *Mask R-CNN* | - | 37.9 | 34.5 |
| *(R50-FPN)* | $\gamma = 80\%$ | **38.5** | 34.5 |

Table 3: Fine-tuning with DropIT on image classification, object detection & instance segmentation.

| Cached | Dropping Rate $\gamma$ | | | | |
|--------|-------------|-------------|-------------|-------------|-------------|
|        | $\gamma = 0\%$ | $\gamma = 60\%$ | $\gamma = 70\%$ | $\gamma = 80\%$ | $\gamma = 90\%$ |
| Tensor | 11.26 G | 4.50 G $(-60\%)$ | 3.38 G $(-70\%)$ | 2.25 G $(-80\%)$ | 1.13 G $(-90\%)$ |

Table 4: Memory cost of DropIT cached tensors (without indices) for different $\gamma$. DropIT can precisely reduce the memory by $\gamma$. The measured model is DeiT-S with a batch size of 1024.

## 4.2 IMPACT ON ACCURACY

**Training from scratch on ImageNet-1k.** Table 2 shows that training DeiT-Ti from scratch without DropIT (baseline) has a top-1 accuracy of 72.1 on ImageNet-1k. Random dropping matches or improves the accuracy (72.4) when $\gamma \leq 20\%$, but with higher $\gamma$ ($\gamma \geq 30\%$), accuracy progressively decreases from the baseline. The phenomenon can be explained by the following: (1) Small amounts of random dropping ($\gamma \leq 20\%$) can be regarded as adding random noise to the gradient. The noise has a regularization effect on the network optimization to improve the accuracy, similar to what was observed in previous studies (Neelakantan et al., 2015; Evans & Aamodt, 2021). (2) Too much random dropping ($\gamma \geq 30\%$) results in deviations that can no longer be seen as small gradient noise, hence reducing performance.

With min-$k$ dropping, DropIT can match or exceed the baseline accuracy over a wide range of $\gamma$ ($\leq 70\%$). Intuitively, training from scratch should be difficult with DropIT, especially under large dropping rates, as the computed gradients are approximations. However, our experiments demonstrate that DropIT achieves 0.4% and 0.3% higher accuracy in $\gamma = 50\%$ and 60%, respectively. In fact, DropIT can match the baseline accuracy even after discarding 70% of the elements.

Fig. 4 compares the loss curves when training from scratch on the baseline DeiT-Ti model without and with DropIT using a min-$k$ strategy. The loss curves of DropIT with various $\gamma$ values follow the same trend as the baseline; up to some value of $\gamma$, the curves are also but are consistently lower than the baseline, with $\gamma = 50\%, 60\%$ achieving the lowest losses and highest accuracies. As such, we conclude that DropIT accurately approximates the gradient while reducing noise, as per our theoretical analysis.

**Fine-tuning on CIFAR-100.** Table 3(a) shows that DeiT networks can be fine-tuned with DropIT to achieve higher than baseline accuracies even while dropping up to 90% intermediate elements. Compared to training from scratch from Table 2, DropIT can work with a more extreme dropping rate (90% vs. 70%). We interpret that this is because the network already has a good initialization before fine-tuning, thereby simplifying the optimization and allowing a higher $\gamma$ to be used.

**Backbone fine-tuning, head network training from scratch, on COCO.** We investigated DropIT in two settings: training from scratch and fine-tuning from a pre-trained network. We also studied a backbone network initialized with ImageNet pre-training, while leaving others, such as RPN and R-CNN head, uninitialized, which is common practice in object detection. Table 3(b) shows that DropIT can steadily improve detection accuracy ($AP^{box}$). When $\gamma = 80\%$, we observed an impressive 0.6 AP gain in Mask R-CNN, although this gain was not observed in $AP^{mask}$. We believe that the segmentation performance may be highly related to the deconvolutional layers in the mask head, which are not currently supported by DropIT. We plan to investigate this further in future work. These experiments demonstrate the effectiveness of DropIT on CNNs, and in Appendix A.4, we demonstrate the effectiveness of DropIT for ResNet training on ImageNet-1k.

| Benchmark | FC Cache | Others Cache | Acc | MaxM | MaxM (-Index) | Speed (ms) |
|---|---|---|---|---|---|---|
| *DeiT-S on CIFAR-100* | *none* | *none* | 89.7 | 6.66 G | 6.66 G | **172** |
| | DropIT ($\gamma = 90\%$) | *none* | **90.1** | **5.59 G** | **5.29 G** | 212 |
| | MESA (8-bit) | MESA (8-bit) | 89.7 | 3.52 G | 3.52 G | 416 |
| | DropIT($\gamma = 90\%$) | MESA (8-bit) | **89.9** | **3.27 G** | **2.97 G** | **375** |
| | GACT (4-bit) | GACT (4-bit) | 89.7 | 2.16 G | 2.16 G | 290+49* |
| | DropIT($\gamma = 90\%$) | GACT (4-bit) | **90.0** | 2.27 G | **1.97 G** | **286+25*** |

*Note: GACT has a time-consuming sensitivity profiling computation every 1000 iterations. It costs 49.81 and 25.43 (+DropIT) seconds in our benchmark. So we add an average of this time over 1000 iterations.*

Table 5: Compare and combine with state-of-the-art ACT methods. FC: fully-connected. MaxM: maximum memory. MaxM (- Index): maximum memory without index (moved to CPU). We follow MESA to use batch size 128 to measure memory and speed on a single GPU.

## 4.3 IMPACT ON MEMORY & SPEED, SOTA COMPARISON

**Intermediate Tensor Cache Reduction.** Table 4 shows the intermediate tensor cache reduction achieved by DropIT. In DropIT applied layers (FC layers of DeiT-S), the total reserved activation (batch size = 1024) is 11.26 G. When we use DropIT to discard activations, the memory reduction is precisely controlled by $\gamma$, *i.e.* $\gamma = 90\%$ means the reduction is $11.26 \times 0.9$. DropIT does incur some memory cost for indexing, but as we show next, the maximum GPU memory can still be reduced.

**Comparison and Combination with SOTA.** In Table 5, we compare and combine DropIT with state-of-the-art activation quantization methods. Measuring performance individually, with $\gamma = 90\%$, DropIT improves accuracy by 0.4 and reduces maximum memory by 1.07 G (1.37G activations - 0.3G indexing), and slightly increases the time (40 ms) per iteration. The max memory reduction is less than that shown Table 4 because activations from non-applicable layers still occupy considerable memory. Therefore, a natural idea to supplement DropIT is to perform activation quantization for layers without DropIT. We next present the combination results of DropIT with recent methods MESA (Pan et al., 2022) and GACT (Liu et al., 2022).

As shown in Table 5, MESA can reduce memory with 8-bit quantization and it has no impact on the baseline accuracy (89.7). However, the time cost of the MESA algorithm is also considerable, and is $416/172 \approx 2.4\times$ slower than baseline and $416/212 \approx 2\times$ more than DropIT, with no accuracy improvement in CIFAR-100 finetuning. MESA achieves 71.9 accuracy of DropIT-Ti on ImageNet-1k, but DropIT can go up to 72.5 (Table 2, $\gamma = 50\%$). We can combine MESA with DropIT by applying DropIT in the `conv`/`fc` layers and applying MESA in the other layers. Together, the accuracy, memory, and speed are all improved over MESA alone, conclusively demonstrating the effectiveness of DropIT.

We compare similarly to GACT; Table 5 shows that at 4 bits, it can reduce max-memory even further. Combining GACT with DropIT marginally increases the max-memory due to DropIT's indexing consumption; however, there are both accuracy and speed gains. Furthermore, GACT reports 0.2~0.4 $AP^{box}$ loss on COCO (Liu et al., 2022), though our DropIT can produce 0.6 $AP^{box}$ improvement on COCO (Table 3(b)). To sum up, DropIT has its unique advantages in terms of accuracy and speed compared to existing activation quantization methods. Although it saves less memory than the latter, we can combine the two to achieve higher memory efficiency.

## 5 CONCLUSION

In this paper, we propose the Dropping Intermediate Tensors (DropIT) method to reduce the GPU memory cost during the training of DNNs. Specifically, DropIT drops elements in intermediate tensors to achieve a memory-efficient tensor cache, and it recovers sparsified tensors from the remaining elements in the backward pass to compute the gradient. Our experiments show that DropIT can improve the accuracies of DNNs and save GPU memory on different backbones and datasets. DropIT provides a new perspective to reduce GPU memory costs during DNN training. For future work, DropIT can be explored in training large (vision-)language models.

## 6 ACKNOWLEDGEMENTS

This research is supported by the National Research Foundation, Singapore under its NRF Fellowship for AI (NRF-NRFFAI1-2019-0001). Any opinions, findings and conclusions or recommendations expressed in this material are those of the author(s) and do not reflect the views of National Research Foundation, Singapore.

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

# A APPENDIX

## A.1 COMPLETE CONVERGENCE ANALYSIS

Here we prove convergence of DropIT with min-$k$ dropping strategy. By scaling the learning rate with a factor of $\frac{1}{\alpha}$, the gradient of min-$k$ dropping is modeled as:

$$g_{min\text{-}k} = g_{gd} + \frac{\beta}{\alpha} n(0, \xi^2). \tag{12}$$

where $n$ is zero-mean noise with a variance of $\xi^2$, and $\alpha$, $\beta$ are varied each iteration.

We assume that the loss function $F$ is $L$-smooth, i.e., $F$ is differentiable and there exists a constant $L > 0$ such that

$$F(y) \leq F(x) + \langle \nabla F(x), y - x \rangle + \frac{L}{2} \|y - x\|^2, \qquad \forall x, y \in \mathbb{R}^d. \tag{13}$$

Performing Taylor expansion we have:

$$\mathbb{E}[F(x_{t+1})] \leq F(x_t) - \langle \nabla F(x_t), x_{t+1} - x_t \rangle + \frac{\eta^2 L}{2} \mathbb{E}[\|\nabla F(x_t)\|^2]$$

$$\leq F(x_t) - \eta \|\nabla F(x_t)\|^2 + \frac{\eta^2 L \xi^2 \beta_t^2}{2\alpha_t^2} \tag{14}$$

Rearranging the terms of the above inequality and dividing by $\frac{\eta}{2}$, we obtain:

$$\|\nabla F(x_t)\|^2 \leq \frac{2(F(x_t) - \mathbb{E}[F(x_{t+1})])}{\eta} + \frac{\eta L \xi^2 \beta_t^2}{\alpha_t^2} \tag{15}$$

Summing up from $t = 1$ to $T$ and divided by $T$, we get:

$$\frac{1}{T} \mathbb{E} \sum_{t=1}^{T} \|\nabla F(x_t)\|^2 \leq \frac{2(F(x_1) - F(x^*))}{T\eta} + \eta L \xi^2 \frac{1}{T} \sum_{t=1}^{T} \frac{\beta_t^2}{\alpha_t^2} \tag{16}$$

where $x^*$ indicates an optimal solution.

## A.2 MODELING MIN-$k$ DROPPING GRADIENT WITH NONLINEAR FUNCTION

We can replace Eq. 8 (gradient model of min-$k$ dropping) with nonlinear function and still achieve the same convergence as in Eq. 10.

The gradient is biased with min-$k$ dropping, we assume it can be modeled as:

$$g_{min\text{-}k} = g_{gd} + \beta n(0, \xi^2) + b, \tag{17}$$

where $b$ is a bias and $\|b\|^2 \leq (1-\alpha)\|g_{gd}\|^2$. $\alpha$ and $\beta$ varies each iteration, i.e., $\alpha = \{\alpha_1, \alpha_2, ..., \alpha_t\}$ and $\beta = \{\beta_1, \beta_2, ..., \beta_t\}$.

Assuming the loss function $F$ is $L$-smooth, we obtain:

$$
\begin{aligned}
\mathbb{E}F(x_{t+1}) &\leq F(x_t) - \langle \nabla F(x_t), x_{t+1} - x_t \rangle + \frac{\eta^2 L}{2}\mathbb{E}||\nabla F(x_t) + b||^2 \\
&= f(x_t) - \eta \langle \nabla F(x_t), \nabla F(x_t) + b \rangle + \frac{\eta^2 L \xi^2 \beta_t^2}{2} \\
&\leq f(x_t) - \eta \langle \nabla F(x_t), \nabla F(x_t) + b \rangle + \frac{\eta}{2}||\nabla F(x_t) + b||^2 + \frac{\eta^2 L \xi^2 \beta_t^2}{2} \\
&= f(x_t) - \frac{\eta}{2}\left( 2\langle \nabla F(x_t), \nabla F(x_t) + b \rangle - ||\nabla F(x_t) + b||^2 \right) + \frac{\eta^2 L \xi^2 \beta_t^2}{2} \\
&= f(x_t) - \frac{\eta}{2}\left( ||\nabla F(x_t)||^2 - ||b||^2 \right) + \frac{\eta^2 L \xi^2 \beta_t^2}{2} \\
&\leq f(x_t) - \frac{\eta \alpha_t}{2}||\nabla F(x_t)||^2 + \frac{\eta^2 L \xi^2 \beta_t^2}{2}
\end{aligned}
\tag{18}
$$

Rearranging the terms of the above inequality and dividing by $\frac{\eta \alpha_t}{2}$, we obtain:

$$
||\nabla F(x_t)||^2 \leq \frac{2(F(x_t) - \mathbb{E}[F(x_{t+1})])}{\eta \alpha_t} + \frac{\eta L \xi^2 \beta_t^2}{\alpha_t}
\tag{19}
$$

Using a learning rate of $\frac{\eta}{\alpha_t}$ instead, we have:

$$
||\nabla F(x_t)||^2 \leq \frac{2(F(x_t) - \mathbb{E}[F(x_{t+1})])}{\eta} + \frac{\eta L \xi^2 \beta_t^2}{\alpha_t^2}
\tag{20}
$$

Summing up from $t = 1$ to $T$ and divided by $T$, we get:

$$
\frac{1}{T}\mathbb{E}\sum_{t=1}^{T}||\nabla F(x_t)||^2 \leq \frac{2(F(x_1) - F(x^*))}{T\eta} + \eta L \xi^2 \frac{1}{T}\sum_{t=1}^{T}\frac{\beta_t^2}{\alpha_t^2}
\tag{21}
$$

where $x^*$ indicates an optimal solution. The convergence is exactly the same as in Appendix A.1.

test

| Optimizer | Learning Rate | Convergence |
|---|---|---|
| SGD | $\eta$ | $\frac{\Delta F}{T\eta} + \eta L \xi^2$ |
| SGD | $\frac{\eta}{\alpha}$ | $\alpha\frac{\Delta F}{T\eta} + \frac{1}{\alpha}\eta L \xi^2$ |
| DropIT | $\frac{\eta}{\alpha}$ | $\frac{\Delta F}{T\eta} + \eta L \xi^2 \frac{1}{T}\sum_{t=1}^{T}\frac{\beta_t^2}{\alpha_t^2}$ |
| DropIT (modeling nonlinearity as in A.2) | $\frac{\eta}{\alpha}$ | $\frac{\Delta F}{T\eta} + \eta L \xi^2 \frac{1}{T}\sum_{t=1}^{T}\frac{\beta_t^2}{\alpha_t^2}$ |

Table 6: Theoretical convergences of SGD and DropIT under $L$-smooth condition

In Table 6, we compare convergence of SGD and DropIT under various learning rates. Under a fixed learning rate, SGD and DropIT differ no both convergence speed (the 1st term in convergence) and error (the 2nd term in convergence formula). For a fair setting, we compare SGD with learning rate $\eta$ and DropIT with learning rate $\frac{\eta}{\alpha}$. With a fixed convergence speed, DropIT theoretically achieves lower error.

## A.3 THEORETICAL ANALYSIS ON $\alpha$ AND $\beta$

In this section we compare the gradients of SGD and DropIT with min-$k$ dropping. Note we slightly change the notation of the gradients from $g_{\text{sgd}}$ and $g_{\text{mink}}$ in the main paper to improve clarity for

element-wise analysis. We denote the gradient of SGD as $G$ and DropIT as $G'$. Both gradients are computed by an input tensor $A$ and intermediate tensor $Z$. In DropIT we drop $\gamma$ percent of the elements in $A$. Thus we have:

$$G = A \times Z \tag{22}$$

$$G' = (A \odot D) \times Z, \tag{23}$$

where $\odot$ is element-wise multiplication and $D$ is a dropping mask where each element is either 1 or 0.

From an element-wise viewpoint, we rewrite the computation of gradients:

$$g_{ij} = \sum_k a_{ik} z_{kj} \tag{24}$$

$$g'_{ij} = \sum_k a_{ik} d_{ik} z_{kj}, \tag{25}$$

where $d_{ik}$ is a mask, i.e., $d_{ik}$ is either 0 or 1 depending on $a_{ik}$.

For the simplicity of analysis, we assume $A$ and $Z$ are independent. Let $\mu$ be the mean value of $A$ and $c$ be the mean of dropped elements, after dropping, $A \odot D$ has a mean of $\mu - c$. Taking expectation over all possible inputs, we have:

$$\begin{aligned} \mathbb{E}[g'_{ij}] &= \mathbb{E}[\alpha g_{ij}] \\ &= \frac{\mu - c}{\mu} \sum_k a_{ik} z_{kj} \\ &= \frac{\mu - c}{\mu} g_{ij}. \end{aligned} \tag{26}$$

Therefore the bias caused by dropping is expected to be $\frac{\mu-c}{\mu}$. Assuming the mean value of $A$ is $\mu$ and the mean of dropped value is $c$, after dropping, $D(A)$ has a mean of $\mu - c$. Thus the bias caused by dropping is $\mathbb{E}[\alpha] = \frac{\mu-c}{\mu}$. Recall that we drop elements with small absolute values. In the extreme case where every element in $A$ has the same value as $\mu$, $c$ will reach the upper bound $\gamma\mu$. Therefore, $\mathbb{E}[\alpha] \geq 1 - \gamma$.

Now we analyze on noise and compute $\beta$. Due to the variation on input samples, we have noise in $A$ and $Z$, which results in noise in $G$ and $G'$. To highlight the noise, we rewrite a noisy element $x$ as $\bar{x} + n_x$, where $\bar{x}$ is the mean value of x and $n_x$ is a zero-mean noise. Applying it to Eq.24 and Eq.25 we arrive at:

$$\bar{g}_{ij} + n_g = \sum_k (\bar{a}_{ik} + n_a)(\bar{z}_{kj} + n_z) \tag{27}$$

$$\bar{g'}_{ij} + n_{g'} = \sum_k d_{ik}(\bar{a}_{ik} + n_a)(\bar{z}_{kj} + n_z). \tag{28}$$

Focusing on the noise of gradients, we obtain:

$$n_g = \sum_k (\bar{a}_{ik} n_z + \bar{z}_{kj} n_a + n_a n_z) \tag{29}$$

$$n_{g'} = \sum_k (d_{ik}\bar{a}_{ik} n_z + d_{ik}\bar{z}_{kj} n_a + d_{ik} n_a n_z). \tag{30}$$

Recall that $d_{ik}$ is a mask depending on $a_{ik}$ and therefore depending on $\bar{a}_{ik}$, thus from Eq.26 we have

$$\mathbb{E}[d_{ik}\bar{a}_{ik}] = \frac{\mu - c}{\mu}\bar{a}_{ik} \approx \bar{a}_{ik} > (1 - \gamma)\bar{a}_{ik}. \tag{31}$$

| Dataset | Method | Top-1 | Top-5 | Memory (GB) |
|---------|--------|-------|-------|-------------|
| CIFAR-100 | ResNet-18 ($32 \times 32$) | 77.96 | 94.05 | 648 |
|  | ResNet-18 ($32 \times 32$) + DropIT ($\gamma = 0.8$) | **78.17** | **94.19** | **598** |
|  | ViT-B/16 ($224 \times 224$) | 90.32 | 98.88 | $20290 \times 4$ |
|  | ViT-B/16 ($224 \times 224$) + DropIT ($\gamma = 0.9$) | **90.90** | **99.02** | **16052**$\times 4$ |
| ImageNet | ResNet-18 ($224 \times 224$) | 69.76 | 89.08 | 2826 |
|  | ResNet-18 ($224 \times 224$) + DropIT ($\gamma = 0.8$) | **69.85** | **89.39** | **2600** |
|  | ViT-B/16 ($224 \times 224$) | 83.40 | 96.96 | $20290 \times 4$ |
|  | ViT-B/16 ($224 \times 224$) + DropIT ($\gamma = 0.9$) | **83.61** | **97.01** | **16056**$\times 4$ |

Table 7: More results of different network architecture achieved by DropIT. ResNet-18 results are training from scratch, and ViT-B/16 are fine-tuning from public ImageNet-21k weights.

Because $\gamma$ percent of $D$ is 0 and the other $1 - \gamma$ percent of $D$ is 1, we obtain $\mathbb{E}[d_{ik}] = 1 - \gamma$. Plugging them in Eq.30 we have:

$$
\begin{aligned}
\mathbb{E}[n_{g'}] &= \mathbb{E}[\beta n_g] \\
&= \sum_k (\frac{\mu - c}{\mu} \bar{a}_{ik} n_z + (1 - \gamma) \bar{z}_{kj} n_a + (1 - \gamma) n_a n_z) \\
&\leq \frac{\mu - c}{\mu} \sum_k (\bar{a}_{ik} n_z + \bar{z}_{kj} n_a + n_a n_z) \\
&= \mathbb{E}[\alpha n_g],
\end{aligned}
\tag{32}
$$

where the inequality is satisfied due to $c \leq \gamma \mu$.

This result tells us $\mathbb{E}[\beta] \leq \mathbb{E}[\alpha]$, which suggests DropIT with min-$k$ dropping has a noise reduction effect and should converge better than SGD.

### A.4 RESNET-50 TRAINING FROM SCRATCH ON IMAGENET-1K

We follow `torchvision` training script to train ResNet with and without DropIT. No hyper-parameters are changed. When $\gamma = 70\%$, ResNet-50 with DropIT achieves 76.3 top-1 accuracy, slightly higher than baseline's 76.1 accuracy, demonstrating the effectiveness of DropIT.

### A.5 MORE NETWORK RESULTS

We present more results of different network architectures as shown in Table 7. ResNet-18 are trained from scratch by 90 epochs on ImageNet, totally following `torchvision` reference code. ViT-B/16 is fine-tuned in 3 epochs from its ImageNet-21k pretrained weights. Our proposed DropIT can improve the accuracy for these setting with lower GPU memory cost.

### A.6 WHY DROPIT IS NOT USED FOR THE NETWORK FIRST & FINAL LAYERS

We do not apply DropIT to `conv/fc` layer if it is the first/final layer of the network. The reason is that this does not save memory:

(1) The first layer: The logic of our DropIT on saving memory can be concluded as: creating a smaller tensor $x_{dropped}$ (*i.e.* by `torch.topk`) from input tensor x, then the input tensor x will be automatically released by python garbage collection. However, popular code style is like:

```
dataloader = ...
loss_func = ...

def model(x):
    x = layer1(x)
    x = layer2(x) # x can be released with DropIT
    ...
    x = layeri1(x) # x can be released with DropIT
    x = layeri(x) # x can be released with DropIT, but cannot save
    maximum memory
```

```
10      return x
11
12  for x, y in dataloader:
13      x = model(x) # x will not be released with DropIT
14      loss_func(x, y).backward()
15      ...
```

As we can see, in the `dataloader` loop, the input `x` to `model` can only be recycled when `model` running is finished. So, using DropIT in `layer1` will not reduce maximum memory — instead, it will increase the maximum memory as DropIT created a new $x_{dropped}$.

(2) The final layer: it is easy to understand that DropIT using in the final layer has no effect on memory. See the code block, when running to `layeri`, the maximum memory should be `layer1` $\sim$ `layeri1` cached tensors plus `x` input to `layeri`. If we use DropIT at `layeri`, then there would be an extra $x_{dropped}$ produced, making the maximum memory even higher.

### A.7   HOW TO SELECT $\gamma$ OF DROPIT

From our experiments, we recommend $\gamma = 70\%$ for training from scratch and $\gamma = 80, 90\%$ for fine-tuning. As DropIT incurs memory cost for indexing, $\gamma$ should be larger than $50\%$ to be meaningful (assuming index data type is `int32` with the same number of bits of `float32` for activation). Empirically, we observe that $\gamma$ is reflected consistently in both training loss and testing accuracy. A too-high $\gamma$ which will bias the gradient will have training losses higher than the baseline. As such, an alternative way to select $\gamma$ is to observe the training loss after a some iterations (*e.g.* 100); if it is lower than the baseline, then the testing accuracy is likely to improve as well.

### A.8   MORE EXPERIMENTAL DETAILS

We list the detailed key training hyper-parameters, though they are totally the same with the offical implementations:

● *DeiT-Ti, training from scratch, ImageNet-1k, w/wo DropIT*[2]: batch size 1024, AdamW optimizer, learning rate $10^{-3}$, weight decay 0.05, cosine LR schedule, 300 epochs, with auto mixed precision (AMP) training;

● *DeiT-S, finetuning from official DeiT-S ImageNet-1k weights, CIFAR-100, w/wo DropIT*[3]: batch size 768, SGD optimizer (momentum 0.9), learning rate $10^{-2}$, weight decay $10^{-4}$, cosine LR schedule, 1000 epochs, with AMP training;

● *DeiT-B, finetuning from official DeiT-B ImageNet-1k weights, CIFAR-100, w/wo DropIT*[4]: batch size 768, SGD optimizer (momentum 0.9), learning rate $10^{-2}$, weight decay $10^{-4}$, cosine LR schedule, 1000 epochs, with AMP training;

● *Faster R-CNN, finetuning from* `torchvision` *ResNet-50 ImageNet-1k weights (V1), COCO, w/wo DropIT*[5]: batch size 16, SGD optimizer (momentum 0.9), learning rate 0.02, weight decay $10^{-4}$, Multistep LR schedule (16,22 epochs), 26 epochs, without AMP training;

● *Mask R-CNN, finetuning from* `torchvision` *ResNet-50 ImageNet-1k weights (V1), COCO, w/wo DropIT*[6]: batch size 16, SGD optimizer (momentum 0.9), learning rate 0.02, weight decay $10^{-4}$, Multistep LR schedule (16,22 epochs), 26 epochs, without AMP training.

---

[2]https://www.github.com/facebookresearch/deit/blob/main/README_deit.md
[3]https://www.github.com/facebookresearch/deit/issues/45
[4]https://www.github.com/facebookresearch/deit/issues/45
[5]https://www.github.com/pytorch/vision/tree/main/references/detection#faster-r-cnn-resnet-50-fpn
[6]https://www.github.com/pytorch/vision/tree/main/references/detection#mask-r-cnn

