# OpenReview forum: "DropIT: Dropping Intermediate Tensors for Memory-Efficient DNN Training"
_ICLR.cc/2023/Conference — ICLR 2023 poster_

### Official Review · Reviewer_zGSK · 2022-10-14

**Confidence:** 3
**Correctness:** 3
**Technical Novelty And Significance:** 3
**Empirical Novelty And Significance:** 3
**Recommendation:** 6

**Clarity, Quality, Novelty And Reproducibility:**

* Clarity and quality. The paper is clearly written and extremely easy to follow.
* Novelty. The paper made clearly novel contribution on top of existing work.
* Reproducibility. The code is uploaded to an anonymous repo for reproducing the results. Although the reviewer does not have bandwidth to carefully check given limited time span of review period, the experiments and theory in this paper look quite clear and consistent.


**Strength And Weaknesses:**

Strength
* The unsurprisingly positive side effect of helping noise reduction in SGD convergence makes this method outstanding, specifically compared with some of other gradient approximation or quantization works, which if not tweaked carefully to balance with memory saving, could lead to inferior performance;
* The proposed approach can be simple enough as a pure python plugin to a PyTorch model (Appendix A.4), which is an advantage to practitioners who wanted to quickly test;
* This approach does not require rematerialization, is orthogonal to quantization, and do not have performance overhead compared with methods like GACT which requires to time-consuming profiling.

Weakness
* Saving the indices, which is an 32bit integer or possibly 64bit on large tensors, could be extra non-trivial storage overhead. The possibility of swapping them to CPU, which as mentioned in the paper indeed helps to mitigate the issue, would incur another overhead of slow transfer between CPU and GPU. Consider an optimistic case where 90% of the elements are pruned (fp32), but needs one extra int32 index, the compression rate becomes 5x. Compared with 4bit quantization which is 8x, it does not have significant advantage; In the worse case, if only 70% of the elements are pruned to retain accuracy, then the compression rate drops to 1.67x, which would be less ideal in general.
* The general tradeoff between training speed vs accuracy is unclear. While Table 5 presents comparison between the proposed method and activation quantization approaches, it is not clearly demonstrated how this approach is compared with other lines of work, for example, rematerialization.
* It’s unclear why the proposed work is only used in FC cache in Table 5. Would be more positive evidence if it could show benefit of pruning other caches while retaining the same level of accuracy. Appendix A.5 provides some general explanation on the inapplicability to a network’s first and last layer, which is completely understandable, but it is not clear how it is connected to Table 5.
* GPU sorting or min-K could be highly non-trivial to optimize beyond using `torch.topk`, and this could possibily incur non-negligible overhead in terms of training speed. The overhead of such operation is not carefully studied in the experiment section.


**Summary Of The Paper:**

This paper introduces a novel idea of reducing memory footprint during training by dropping elements in the intermediate tensors whose absolute values are closest to zero.

This paper makes clear contribution in terms of the following aspects:
* proposed a novel approach that is different than gradient checkpointing which requires recomputation, activation compression which uses quantization rather than sparsification or gradient approximation which has a pre-define fixed threshold.
* induces a natural noise reduction that could benefit SGD convergence, which is theoretically proven and justified by experiments;
* demonstrates the effectiveness of the approach with comprehensive experiments, which shows that in most of the cases, most (70% - 90%) of the elements in the intermediate tensor could be dropped with even better accuracy.


**Summary Of The Review:**

This paper is clearly written with clear novelty and contribution. The reviewer believes that its strength outweighs its limitation, and would love to get some clarity in rebuttal, specifically on weakness 1, 2 and 3.

---

> ### Author Response · Authors · 2022-11-09
> **Response to Reviewer zGSK.**
>
> >*Weakness 1: Saving the indices, which is an 32bit integer or possibly 64bit on large tensors, could be extra non-trivial storage overhead. The possibility of swapping them to CPU, which as mentioned in the paper indeed helps to mitigate the issue, would incur another overhead of slow transfer between CPU and GPU. Consider an optimistic case where 90% of the elements are pruned (fp32), but needs one extra int32 index, the compression rate becomes 5x. Compared with 4bit quantization which is 8x, it does not have significant advantage; In the worse case, if only 70% of the elements are pruned to retain accuracy, then the compression rate drops to 1.67x, which would be less ideal in general.*
>
> You're right.  Our compression rate will suffer in the extreme quantization case.  As future work, we are working on an approach that uses block-based sampling, which will greatly reduce indexing overhead. At the same time, 32-bit and 64-bit are still the most commonly used training settings nowadays, and we provide improvements for them.
>
> >*Weakness 2: The general tradeoff between training speed vs accuracy is unclear. While Table 5 presents comparison between the proposed method and activation quantization approaches, it is not clearly demonstrated how this approach is compared with other lines of work, for example, rematerialization.*
>
>
> We did not compare top-k activation sparsification to rematerialization because our goal is to provide a new orthogonal opportunity for activation compression training (ACT), as well as theoretical insight into why top-k activation sparsification will improve performance. In the updated version, we are happy to include an investigation into the combination of rematerialization and DropIT.
>
> >*Weakness 3: It’s unclear why the proposed work is only used in FC cache in Table 5. Would be more positive evidence if it could show benefit of pruning other caches while retaining the same level of accuracy. Appendix A.5 provides some general explanation on the inapplicability to a network’s first and last layer, which is completely understandable, but it is not clear how it is connected to Table 5.*
>
> For some layers, e.g. batch normalization and activations, the partial $\frac{\partial l(a, \theta)}{\partial a}$ may also depend on $a$. In these cases, we do not drop the cache of intermediate tensors as this will affect subsequent back-propagation, causing an aggregated dropping effect on the gradient and will lead to compromised accuracy. Moreover, our focus is on the combination of DropIT with existing ACT methods. By applying DropIT to FC layers and the existing ACT methods to other layers, we achieve better accuracy, higher memory reduction, and shorter training times. (In Table 5)
>
>
>
> >*Weakness 4: GPU sorting or min-K could be highly non-trivial to optimize beyond using torch.topk, and this could possibily incur non-negligible overhead in terms of training speed. The overhead of such operation is not carefully studied in the experiment section.*
>
> You are totally correct. In our early exploration, we proposed optimized parallel top-k methods, which largely reduced overhead. However, in the most recent update, "torch.topk" adapted a new parallel optimized version and achieves similar speed performance to our optimized top-k. So we replace our parallel top-k with  "torch.topk" and no longer include it.
> The overhead of top-k and parallel top-k is shown in the following table:
> |                | γ = 0.5 | γ = 0.6 | γ = 0.7 | γ = 0.8 | γ = 0.9 | γ = 0.99 | Baseline |
> |-------------|---------|---------|---------|---------|---------|----------|----------|
> | top-k         | 898 M   | 840 M   | 710 M   | 602 M   | 512 M   | 434 M    | 646 M    |
> |                | 695 ms  | 798 ms  | 807 ms  | 813 ms  | 820 ms  | 810 ms   | 62 ms    |
> | parallel top-k | 898 M   | 812 M   | 710 M   | 598 M   | 510 M   | 434 M    | 646 M    |
> |                | 70 ms   | 69 ms   | 68 ms   | 66 ms   | 65 ms   | 63 ms    | 62 ms    |
> Reported results are top-1 accuracy, memory cost, and training speed per iteration of ResNet18-DropIT on CIFAR-100.
>
> In the top-k strategy, the smallest $\gamma$ fraction of elements are selected, which produces $O(BCHW)$ time complexity by running a standard top-k algorithm. Parallel top-k processs every $D$ ($D \ll BCHW$) elements in parallel, decreasing the time complexity to $O(D)$. we chose $D=2^{14}$ in our experiments.

---

### Official Review · Reviewer_xcyW · 2022-10-24

**Confidence:** 4
**Correctness:** 2
**Technical Novelty And Significance:** 2
**Empirical Novelty And Significance:** 3
**Recommendation:** 6

**Clarity, Quality, Novelty And Reproducibility:**

I raised most of my concerns in the Weakness part.

The quality of the theoretical analysis is worrying as I have pointed out. I think the approximation and convergence analysis are a bit stretched. The paper also ignores a good number of prior work of different forms of Dropouts, which are directly connected to this proposed idea.

On the novelty side, the paper also claims that they are the first to look at the activation sparsification problem. I would suggest many alternative Dropout papers are attempting this same problem but probably from a different angle. The training efficiency perspective on this problem is novel, but not really any of these random or min-k based dropping methods.

The authors also mainly focused on theoretical performance gains instead of the real implications of the proposed method. You can not simply say fine-grained sparsity of this can directly translate to hardware benefits. There are generally overheads in this. For instance, you might want to reduce the storage using index-based storage or certain encoding schemes. But this also means there is an overhead of encoding and decoding, the authors simply ignored these factors, and I suggest this makes the reported performance gains unrealistic.

**Strength And Weaknesses:**

Strength

- The paper is well-written and easy to understand. The graphical illustration really helps build understanding of the proposed technique.
- The evaluation is on large scale vision datasets (eg. ImageNet and COCO), also vision transformers are included.

Weakness

- A large volume literatures on Dropout are ignored by the authors? I would think this line of work is directly related to this paper.
- I am not comfortable with the theoretical analysis shown in Section 3.4.
    - $g_{min-k} = \alpha g_{gd} + \beta n(0, \sigma)$ (Equation 8) does not look like a good approximation to your min-k operation. Simply speaking, min-k is non-linear and I do not really understand how this linear function can approximate that well?
    - The convergence proof is confusing. Surely your bound is tighter if you use a learning rate that is scaled by $\frac{1}{\alpha}$? Like this does make too much sense, if I say SGD has a smaller LR and its LR scale is greater than my noise scale, it is surely tighter under the L-smooth condition. What is the actual point of showing this? Do you consider this as a proof?
- The theoretical gain is not achievable.
    - The results in Table3 is confusing. In my opinion, the authors are saving ‘memory bandwidth’ but not really using smaller GPU memory. Are you sure your forward activation values are kept on-chip (in SRAM or Cache) in between the feedforward and backward computation? I thought the GPU model will have to push these values into DRAM after finishing one layer of forward computation and load them back from DRAM for backward computation? If this is the case, the proposed technique does not optimize for SRAM space, but in fact is optimizing for the DRAM bandwidth. I will say this is an equally important topic for efficiency since most training workloads are DRAM-bound, but I do not think this paper is making a clear discussion on this.
    - ‘the memory reduction is precisely controlled by γ,
    *i.e*. γ = 90% means the reduction is 11.26×0.9’ → I simply disagree with this statement and do not think this is achievable with out custom silicon.

**Summary Of The Paper:**

This paper proposes a strategy, named DropIT, for activation compression during training. The proposed method can be applied with random dropping or min-k based dropping. The authors show that DropIT can also reduce the gradient noise during training, and provided empirical evidence that the proposed method can drop up to 90% intermediate tensor elements on FC and convolutional layers.

**Summary Of The Review:**

Given the concerns on the theoretical analysis and novelty of this paper, I do not think it is ready to be accpeted to ICLR at its current format.

---

> ### Author Response · Authors · 2022-11-09
> **Response to Reviewer xcyW (2)**
>
> >*(Equation 8) does not look like a good approximation to your min-k operation. Simply speaking, min-k is non-linear and I do not really understand how this linear function can approximate that well?*
>
> In equation 22 of appendix A.2, we show that $E[g_{min-k}] = \frac{\mu-c}{\mu} E[g_{gd}]$, where a linear relationship is implied.
>  Alternatively, one can model the gradient with non-linear function following [1]: $g_{min-k}=g_{gd}+n(0,\xi^2)+b$, where $b$ is a bias and $||b||^2<=(1-\alpha)||g_{gd}||^2$. Here $b$ captures bias while $\beta$ captures the change on variance respectively. (if $||b||^2>||g_{gd}||^2$, bias dominates and it suggests our machine learning model may never converge, which is in contradiction with experiments. Therefore it is reasonable to assume  $||b||^2<=(1-\alpha)||g_{gd}||^2$).
>
> For the purpose of simplicity, we set $\Delta F = 2(F(x_1)-F(x^*))$.
> Based on Lemma 3 of [1], the convergence of DropIT is bounded by $\frac{\Delta F}{T\eta}+\eta L \xi^2 \frac{1}{T} \sum_{t=1}^T\frac{\beta_t^2}{\alpha_t^2}$. Under the non-linear modeling, our discussion on $\alpha$, $\beta$ and conclusions still hold. $E[\alpha]>=E[\beta]$ holds as well.
>
> [1] Ajalloeian, Ahmad, and Sebastian U. Stich. "On the convergence of SGD with biased gradients." arXiv preprint arXiv:2008.00051 (2020).
>
> >*The convergence proof is confusing. Surely your bound is tighter if you use a learning rate that is scaled by 1/a? Like this does make too much sense, if I say SGD has a smaller LR and its LR scale is greater than my noise scale, it is surely tighter under the L-smooth condition. What is the actual point of showing this? Do you consider this as a proof?*
>
> | Optimizer | Learning Rate | Convergence under L-smooth Condition|
> | --- | ----------- |------------------------------------|
> | SGD | $\eta$ |$\frac{\Delta F}{T\eta}+\eta L \xi^2$|
> | SGD | $\frac{\eta}{\alpha}$ | $\alpha \frac{\Delta F}{T\eta}+ \frac{1}{\alpha} \eta L \xi^2$|
> |DropIT| $\frac{\eta}{\alpha}$| $\frac{\Delta F}{T\eta}+\eta L \xi^2 \frac{1}{T} \sum_{t=1}^T\frac{\beta_t^2}{\alpha_t^2}$  *|
> |DropIT (modeling gradient with non-linear function)| $\frac{\eta}{\alpha}$ |$\frac{\Delta F}{T\eta}+\eta L \xi^2 \frac{1}{T} \sum_{t=1}^T\frac{\beta_t^2}{\alpha_t^2}$ |
> * $\frac{1}{T}$ and squares are missing in our submission. We will revise it in a new version.
>
> Learning rate controls the trade-off between convergence speed and error. Scaling learning rate by $\frac{1}{\alpha}$ not only affects the error (the second term in convergence), but also affects convergence speed (the first term).
> If we compare SGD and DropIT under a fixed learning rate, they differ on both convergence speed and error (row 2 and row 3 in the above table), and it is hard to tell which optimizer is better.
>
> For a fair comparison, we compared SGD with learning rate $\eta$ and DropIT with learning rate $\frac{\eta}{\alpha}$ (row 1 and row 3 in the above table) instead, as they have the same convergence speed. We found that under the same convergence speed, DropIT achieves lower error.

---

> > ### Comment · Reviewer_xcyW · 2022-11-18
> > **I raised my score**
> >
> > After reading the author's reply message, I think some of my major concerns are addressed and I have moved my score to a 6.
> >
> > I actually find the table the authors provided for the LR and their corresponding convergence under L-smooth condition fairly interesting. The authors have also addressed my concern on Table 3.
> >
> > The reason why I cannot vote for a full accept for this paper is probably in line with some other reviewers. The idea of gradient stashing may not be super novel.

---

> > > ### Author Response · Authors · 2022-11-24
> > > **Thank you for your positive feedback.**
> > >
> > > Thank you for your positive feedback and for recognizing our contribution on the theoretical analysis.
> > >
> > > In terms of technical novelty, we would like to provide a discussion of the comparison with a similar approach “GIST” (mentioned by Reviewer GQ3x).
> > >
> > > GIST performs "sparse storage and dense compute" on the RELU-Conv feature map, based on the observation that ReLU outputs have high sparsity.
> > >
> > > We want to highlight the following differences between GIST and DropIT:
> > >
> > > 1. GIST performs "lossless" sparsification on the feature map, and the dropping rate depends on the sparsity (number of zeros on the original feature map), whereas DropIT is lossy and does not require high sparsity on the original feature map.
> > > 2. GIST is only applicable to the ReLU-Conv Link in the CNN. DropIT can be applied to all Conv layers in CNN, and FC layers, making it suitable for transformers (GeLU is used as activation function)
> > > 3. GIST will not affect the training process as it performs lossless sparsification. Despite lossy sparsification, DropIT can provide benefits to training and result in higher accuracy. We focus on the analysis of this phenomenon.
> > >
> > >
> > > If you have any additional questions, we would be happy to provide additional explanations.
> > > Thank you very much.

---

> ### Author Response · Authors · 2022-11-09
> **Response to Reviewer xcyW (1)**
>
> >*A large volume literatures on Dropout are ignored by the authors? I would think this line of work is directly related to this paper.*
>
> Thanks for your suggestion, our work falls into the line of work that only touches the backwards pass, e.g. ACT[1], which we think is a closer discussion. Dropout modified the forward pass, and the objective is not memory-efficient training. We are happy to add a discussion on dropout to the revision as the idea of "dropping" is similar.
>
> [1]Xiaoxuan Liu et al. GACT: Activation compressed training for generic network architectures. In ICML, pp. 14139–14152, 2022.
>
> >*The results in Table3 is confusing. In my opinion, the authors are saving ‘memory bandwidth’ but not really using smaller GPU memory. Are you sure your forward activation values are kept on-chip (in SRAM or Cache) in between the feedforward and backward computation? I thought the GPU model will have to push these values into DRAM after finishing one layer of forward computation and load them back from DRAM for backward computation? If this is the case, the proposed technique does not optimize for SRAM space, but in fact is optimizing for the DRAM bandwidth. I will say this is an equally important topic for efficiency since most training workloads are DRAM-bound, but I do not think this paper is making a clear discussion on this.*
>
> Thank you for bringing this to our attention. We've noticed some confusion about the term "cache" in the fields of deep learning and computer architecture. In this context, "the tensor is cached" means that the tensor will not be deleted from DRAM after completing the forward pass, because it is required for the backward pass gradient computation. The tensor will be stored in GPU DRAM for the duration of its life. Despite adhering to previous work, we believe this term is unclear and will revise it in the updated version.
>
>
>
> >*the memory reduction is precisely controlled by γ, i.e. γ = 90% means the reduction is 11.26×0.9’ → I simply disagree with this statement and do not think this is achievable with out custom silicon.*
>
> Following the preceding discussion, "cached tensor" does not imply "real caching" in this context. In comparison to the baseline, we remove 90% of the stored intermediate tensors from the GPU memory, resulting in a 90% reduction in DRAM memory consumption.
>
>
>
> >*On the novelty side, the paper also claims that they are the first to look at the activation sparsification problem. I would suggest many alternative Dropout papers are attempting this same problem but probably from a different angle. The training efficiency perspective on this problem is novel, but not really any of these random or min-k based dropping methods.
> The authors also mainly focused on theoretical performance gains instead of the real implications of the proposed method. You can not simply say fine-grained sparsity of this can directly translate to hardware benefits. There are generally overheads in this. For instance, you might want to reduce the storage using index-based storage or certain encoding schemes. But this also means there is an overhead of encoding and decoding, the authors simply ignored these factors, and I suggest this makes the reported performance gains unrealistic.*
>
> Thank you for pointing this out to us. The accuracy and memory evaluations were carried out on the RTX-A5000 GPU. We already use index-based storage for sparsified tensors, as introduced in the method section. The time overhead for encoding and decoding is already included in the Table.5. We've also included our source code so the reader can verify the numbers on their own device.

---

### Official Review · Reviewer_Ltqg · 2022-10-25

**Confidence:** 5
**Correctness:** 3
**Technical Novelty And Significance:** 3
**Empirical Novelty And Significance:** 3
**Recommendation:** 5

**Clarity, Quality, Novelty And Reproducibility:**

I believe the contribution of this paper is novel and clear. The code has been provided which makes the results reproducible although I haven't tested the code myself.


**Strength And Weaknesses:**

(Strengths:) In general, I believe the idea of reducing the memory usage of neural networks during training/fine-tuning is of paramount importance specially for transformer models. This work is an attempt to address this issue with a novel idea. The paper is also well-written and easy to understand.

(Weaknesses:) However, I am not convinced about the effectiveness of the proposed method due to lack of sufficient experiments. More precisely, the choice of DeiT and fine-tuning it for 1000 epochs doesn't reflect the effectiveness of the proposed method. Instead, I believe ViT models should be used for fine-tuning which requires up to 4 epochs. Moreover, I expected to see more CNNs tested such as ResNet-18/MobileNet on CIFAR and ImageNet as an example.


**Summary Of The Paper:**

This paper presents a method that reduces the memory usage of training/fine-tuning for convolutional neural networks and vision transformers. The idea is to prune away insignificant intermediate tensors (i.e., those that are low in value) required for the back-propagation. The method has been tested on DeiT and R-CNN models. It has been shown that up to 90% of intermediate tensors can be pruned away without any performance degradation. However, this reduction doesn't linearly translate into memory reduction. For example, it was shown that with sparsity rate of 90%, there is only about 1 GB memory reduction when fine-tuning DeiT on CIFAR-100.

**Summary Of The Review:**

In my opinion, the contribution of this paper is novel and is addressing a real challenge which memory usage reduction during fine-tuning. However, the proposed method was not fully tested which makes me wonder about its generality. For the rebuttal, I would like to see how this method works when fine-tuning ViT-base model on CIFAR-10 and CIFAR-100 for 3 to 4 epochs, which is a common practice. It would be great to include some results from BERT on GLUE and SQuAD tasks too (again when fine-tuning it for a couple of epochs). I would also like to see some results for small-size CNNs such as MobileNet on ImageNet-1K or even ResNet-18 on CIFAR-10. I am also wondering why there is only 1GB memory reduction on DeiT when gamma is 90% in Table 5; how much of the memory usage is dedicated for the intermediate tensors out of 6.66GB in this case?

---

> ### Author Response · Authors · 2022-11-09
> **Response to Reviewer Ltqg**
>
> >*However, I am not convinced about the effectiveness of the proposed method due to lack of sufficient experiments. More precisely, the choice of DeiT and fine-tuning it for 1000 epochs doesn't reflect the effectiveness of the proposed method. Instead, I believe ViT models should be used for fine-tuning which requires up to 4 epochs. Moreover, I expected to see more CNNs tested such as ResNet-18/MobileNet on CIFAR and ImageNet as an example.*
>
>
> Thanks for your suggestions. For DeiT, we follow the fine-tuning instructions in the original paper (https://github.com/facebookresearch/deit/issues/45#issuecomment-765213622). In addition, we do have results on ResNet18 training from scratch on ImageNet, and the ViT model for finetuning in 3 epochs in our early exploration.
>
> | Dataset   | Method                                 | Top-1 | Top-5 | Memory (MB) |
> |-----------|----------------------------------------|-------|-------|-------------|
> | CIFAR-100 | ResNet-18 (32x32)                      | 77.96 | 94.05 | 648         |
> |           | ResNet-18 (322) + DropIT (γ = 0.8)     | 78.17 | 94.19 | 598         |
> |           | ViT-B/16 (224x224)                     | 90.32 | 98.88 | 20290×4     |
> |           | ViT-B/16 (224x224) + DropIT (γ = 0.9)  | 90.90 | 99.02 | 16052×4     |
> | ImageNet  | ResNet-18 (224x224)                    | 69.76 | 89.08 | 2826        |
> |           | ResNet-18 (224x224) + DropIT (γ = 0.8) | 69.85 | 89.39 | 2600        |
> |           | ViT-B/16 (224x224)                     | 83.40 | 96.96 | 20290×4     |
> |           | ViT-B/16 (224x224) + DropIT (γ = 0.9)  | 83.61 | 97.01 | 16056×4     |
>
> where DropIT achieves higher accuracy than baseline with less memory consumption on different backbone networks and datasets."×4" means training on 4 GPUs.We will include the training details in the future revision.
>
>
> For BERT, we do not have time to experiment, so we will not claim that our method works for it but we will experiment in the updated version.
>
> >*I am also wondering why there is only 1GB memory reduction on DeiT when gamma is 90% in Table 5; how much of the memory usage is dedicated for the intermediate tensors out of 6.66GB in this case?*
>
> As we only apply DropIT to the FC layer, the overall memory reduction ratio is amortized as the intermediate tensors for other layers (e.g., GeLU, Softmax) are untouched. However, our focus is on the combination of DropIT with existing ACT methods. By applying DropIT to FC layers and the existing ACT methods to other layers, we achieve better accuracy, higher memory reduction, and shorter training times (in Table 5).

---

> > ### Comment · Reviewer_Ltqg · 2022-12-06
> > **Re: Response to Reviewer Ltqg**
> >
> > Thank you for your comments. According to the results of ViT, it seems that the effectiveness of the proposed method is dependent of batch size in memory reduction. The structure of ViT is similar to DeiT, however, the results provided in terms of memory usage are different; my guess is the batch size; I might be wrong but there is no explanation in the paper. There is another interesting point about ViT and ResNet. I understand why memory reduction for ViT and DeiT is marginal (which is the main weakness of the paper) when using the proposed method. However, it shouldn't be the case for ResNet as I expected to see more reductions. Therefore, I agree with other reviewers that this paper lies on the borderline and I'll keep my score.

---

### Official Review · Reviewer_GQ3x · 2022-10-31

**Confidence:** 4
**Correctness:** 3
**Technical Novelty And Significance:** 2
**Empirical Novelty And Significance:** 3
**Recommendation:** 6

**Clarity, Quality, Novelty And Reproducibility:**

The authors give some details on how their pytoch implementation works, and combined with the simplicity of the technique it should not be hard to reproduce the results. The paper is also clear and well-written.

The theoretical proof seems to be based on shaky assumptions. I'm not an expert in this area so I would like to see the authors' response.

The novelty of the idea does not seem very high. I already mentioned the GIST paper and there may be other papers which study pruning on stashed activations.

**Strength And Weaknesses:**

Strenghts:
 - Straightforward and easy to implement method that seems fairly robust, at least on vision benchmarks

Weaknesses:
 - The idea is not novel: it's already part of the GIST paper. GIST combines Min-K stashed activation compression with binarized stashed ReLU masks. The GIST paper is systems-focused and contains multiple techniques. This paper gives a more focused study of the Min-K sparsity technique and is still a useful result.
 - The memory savings is great but the actual run time savings seem unimpressive. Perhaps it would be better to focus on benchmarks which are much more memory-hungry such as Transformers. These models are typically trained with microbatching and increasing the microbatch size should grant a big performance boost.

Question:
 - The proof that DropIT can decrease the gradient noise depends on Equation 8, where the noise introduced by the Min-K sparsification is modeled as $g_{min-k} = \alpha g_{gd} + \beta n(0, \xi^2)$. It's not clear to me why this noise is zero-mean. In CNNs, we have non-symmetric activation functions like ReLU. Min-K sparsity would change the means of the stashed activations tensors, which in turn would change the means of the gradient update tensors. I am not an expert here so I would like to see the authors' response.

[1]https://ieeexplore.ieee.org/abstract/document/8416872?casa_token=0XtaIex9huQAAAAA:RwuzO_eOgBsEms6Y3XESj66oOFSaihBdAT0vVXlq_YETd7xlO_ifa9feFTDT0wrGlU1KKM7D

**Summary Of The Paper:**

The authors propose DropIT, where stashed intermediate tensors are pruned via a Top-K function and converted to a compressed sparse format. This sparsification reduces the memory consumption of stashing, enabling larger batch sizes and faster training. During the backward pass, the stashed tensors are first decompressed to dense representation since the sparsity is usually not high enough for sparse matmuls to be effective.

The authors show that by scaling the learning rate, DropIT can actually decrease the gradient noise.

Experiments show that a variety of vision benchmarks such as DeiT and R-CNN can be trained on datasets like ImageNet and COCO with 70%-90% sparsity without reducing the final accuracy. This is a great result. However, wall-clock time improvements seem less impressive, without only about 10% run time improvement in Table 5.

EDIT: raised score from 5 to 6 after rebuttal

**Summary Of The Review:**

DropIT proposes Min-K pruning of stashed activations to save memory. This isn't a novel idea and has been investigated in at least the GIST paper[1]. However, this paper is still useful as it presents an isolated study on this technique and presents impressive results on memory savings for vision benchmarks.

I would raise the score if I see additional discussion regarding the assumptions for the theoretical proof and results on more memory-intensive benchmarks.

[1] https://ieeexplore.ieee.org/abstract/document/8416872?casa_token=0XtaIex9huQAAAAA:RwuzO_eOgBsEms6Y3XESj66oOFSaihBdAT0vVXlq_YETd7xlO_ifa9feFTDT0wrGlU1KKM7D

---

> ### Author Response · Authors · 2022-11-09
> **Response to Reviewer GQ3x**
>
> >*The idea is not novel; it's already part of the GIST paper. GIST combines Min-K stashed activation compression with binarized stashed ReLU masks. The GIST paper is systems-focused and contains multiple techniques. This paper gives a more focused study of the Min-K sparsity technique and is still a useful result.*
>
> Thanks for pointing this out. We did cite GIST, but we could not find where they used min-K. Regardless, using min-K for sparsification is indeed common in the field (e.g., distributed training [1]). One of our aims is to provide theoretical insight into why min-K sparsification improves training and we thank you for acknowledging this contribution.
>
> [1]Alham Fikri Aji and Kenneth Heafield. Sparse communication for distributed gradient descent. In EMNLP, pp. 440–445, 2017.
>
>
>
> >*The memory savings are great but the actual run time savings seem unimpressive. Perhaps it would be better to focus on benchmarks which are much more memory-hungry such as Transformers. These models are typically trained with microbatching and increasing the microbatch size should grant a big performance boost.*
>
> As our aim is to provide memory reduction and higher accuracy, saving run time is not our focus. Nonetheless, previous works on activation compression training all result in increased time consumption because they basically trade off time and accuracy for memory, whereas we achieve a better trade-off on time and higher accuracy, as shown in Table 5. In these comparisons, we keep our batch size the same as the original works so that we can make a fair comparison to draw conclusions on the theoretical improvement, and increasing the batch size complicates analysis.
>
> We appreciate your advice on benchmarks that are much more memory-hungry, such as Transformers. It would be great if you could help us provide more details on what types of transformers are expected (as we already provide results for the Vision Transformer).
> For micro-batch training, increasing the batch size will reduce the iterations and thus the training time, and our approach applies with this conclusion. The reduction in time will be proportional to the increase in batchsize, which can be derived from the ratio of the memory reduction.
>
> >*It's not clear to me why this noise is zero-mean. In CNNs, we have non-symmetric activation functions like ReLU. Min-K sparsity would change the means of the stashed activations tensors, which in turn would change the means of the gradient update tensors. I am not an expert here so I would like to see the authors' response.*
>
> You are correct that min-k sparsity would change the means of the gradient update tensors.  In fact, min-k would change both the mean and variance.  The mean could be non-zero. Without affecting convergence, we model it as zero mean while migrating non-zero mean to bias (captured by $\alpha$).  In equation 8, we have $g_{min-k}= \alpha g_{gd} + \beta n(0,\xi^2)$, where $\alpha$ captures the change on mean while $\beta$ captures the change on variance respectively. If there is a noise with non-zero mean, we can always migrate the non-zero mean to $\alpha$, then set the noise to be zero-mean. Zero-mean noise setting is also used in analyzing top-k gradient compression and random-k compression [2]. By doing so, the analysis can be simplified without affecting convergence.
>
> [2] Ajalloeian, Ahmad, and Sebastian U. Stich. "On the convergence of SGD with biased gradients." arXiv preprint arXiv:2008.00051 (2020).

---

> > ### Author Response · Authors · 2022-11-24
> > **Revision on the comment.**
> >
> > We would like to provide a revision on the comment:
> >
> > ***1. comparision with GIST.***
> >
> >
> >
> > GIST performs "sparse storage and dense compute" on the RELU-Conv feature map, based on the observation that ReLU outputs have high sparsity.
> >
> > We would like to highlight the following differences between GIST and DropIT:
> >
> > 1. GIST performs "lossless" sparsification on the feature map, and the dropping rate depends on the sparsity (number of zeros on the original feature map), whereas DropIT is lossy and does not require high sparsity on the original feature map.
> > 2. GIST is only applicable to the ReLU-Conv Link in the CNN. DropIT can be applied to all Conv layers in CNN, and FC layers, making it suitable for transformers (GeLU is used as activation function)
> > 3. GIST will not affect the training process as it performs lossless sparsification. Despite lossy sparsification, DropIT can provide benefits to training and result in higher accuracy. We focus on the analysis of this phenomenon.
> >
> >
> > ***2. Additional discussion on memory-hungry benchmarks:***
> >
> > In the experiments for full-precision training, our setting is Deit-S with a batch size of 1024 for 4 GPUs; the total memory cost is ~40G. We provide here the finetuning result for ViT models, where the total memory cost is nearly 80G.
> >
> > | Dataset   | Method                                 | Top-1 | Top-5 | Memory (MB) |
> > |-----------|----------------------------------------|-------|-------|-------------|
> > | CIFAR-100  | ViT-B/16 (224x224)                     | 90.32 | 98.88 | 20290×4     |
> > |           | ViT-B/16 (224x224) + DropIT (γ = 0.9)  | 90.90 | 99.02 | 16052×4     |
> > | ImageNet   | ViT-B/16 (224x224)                     | 83.40 | 96.96 | 20290×4     |
> > |           | ViT-B/16 (224x224) + DropIT (γ = 0.9)  | 83.61 | 97.01 | 16056×4     |

---

> > > ### Comment · Reviewer_GQ3x · 2022-11-28
> > > **Raising score to a 6**
> > >
> > > The authors have addressed most of my doubts and I think the paper is a 6 (borderline accept). I do think this DropIT a useful technique and the evaluation looks solid. But DropIT feels like an incremental improvement over existing techniques to save on stashing memory.

---

### Decision · Program_Chairs · 2023-01-20

**Decision:**

Accept: poster

**Justification For Why Not Higher Score:**

After rebuttal, the more experiments are conducted which gives more convincing evaluations. But it seems that the effectiveness of the proposed method is dependent of batch size in memory reduction.

**Justification For Why Not Lower Score:**

The paper is clearly written with novel idea, supported by several experiments on large scale dataset (eg. ImageNet and COCO), also vision transformers are included.

**Metareview: Summary, Strengths And Weaknesses:**

- Summary
This paper presents a method that reduces the memory usage of training/fine-tuning for convolutional neural networks and vision transformers. The idea is to prune away insignificant intermediate tensors required for the back-propagation. It has been shown that up to 90% of intermediate tensors can be pruned away without any performance degradation. DropIT can also reduce the gradient noise during training. The paper is well-written and easy to understand.

- Strengths
In general, the idea of reducing the memory usage of neural networks during training/fine-tuning is of paramount importance specially for transformer models. This work is an attempt to address this issue with a novel idea.

- Weakness
The effectiveness of the proposed method is not very convincing due to insufficient experiments.
DropIT seems to be an incremental improvement over existing techniques to save on stashing memory.

**Note From Pc:**

if the above contains the word "oral" or "spotlight" please see: "oral" presentation means -> notable-top-5% and "spotlight" means -> notable-top-25%. As stated in our emails, we are disassociating presentation type from AC recommendations